# Dimensional Stability and Reproducibility of Varying FFF Models for Aligners in Comparison to Plaster Models

**DOI:** 10.3390/ma16134835

**Published:** 2023-07-05

**Authors:** Nina Lümkemann, Melisa Klimenta, Moritz Hoffmann, John Meinen, Bogna Stawarczyk

**Affiliations:** Department of Prosthetic Dentistry, LMU University Hospital, LMU Munich, 80336 Munich, Germany; nina.luemkemann@med.uni-muenchen.de (N.L.); melisa.klimenta@med.uni-muenchen.de (M.K.); moritz.hoffmann@med.uni-muenchen.de (M.H.); john.meinen@med.uni-muenchen.de (J.M.)

**Keywords:** additive manufacturing, models, FFF, aligner, thermoforming, dimensional stability

## Abstract

To test the impact of FFF filaments, printing parameters, thermoforming foils, repeated thermoforming cycles, and type of jaw on the dimensional stability of FFF models for aligners and to compare them with plaster models, FFF models (maxilla, n = 48; mandible, n = 48) from two filaments (SIMPLEX aligner and Renfert PLA HT, both Renfert GmbH) were fabricated using four printing parameters (one, two, or three loops; four loops acted as the default) and conventional plaster models (n = 12) based on a young, female dentition. All models were thermoformed under pressure three times in total using two different thermoforming foils, namely 0.75 mm × 125 mm Ø aligner foil (CA Pro+ Clear Aligner, Scheu Dental) and 1.0 mm × 125 mm Ø Duran foil (Duran+, Scheu Dental). Aligner foil was heated at 220 °C for 25 s and Duran foil at 220 °C for 30 s. All models were scanned after fabrication as well as after each thermoforming cycle. The obtained STL datasets were analyzed using the local best-fit method (GOM Inspect Pro, Carl Zeiss Metrology GmbH). Data were analyzed using a Kolmogorov–Smirnov-test, a one-way ANOVA with post-hoc Scheffé, and a *t*-test (*p* < 0.05). The dimensional stability of the models was most strongly affected by the printing parameters (number of loops; η_p_^2^ = 0.768, *p* < 0.001) followed by the thermoforming foil used (η_p_^2^ = 0.663, *p* < 0.001) as well as the type of model (η_p_^2^ = 0.588, *p* < 0.001). In addition, various interactions showed an influence on the dimensional stability (η_p_^2^ = 0.041–0.386, *p* < 0.035). SIMPLEX maxillary models (default; four loops), thermoformed using aligner foil, showed higher deformation stability than did plaster models. These initial FFF models provide comparable precision to plaster models, but the dimensional stability of the FFF models, in contrast to that of plaster models, decreases with increasing numbers of thermoforming cycles.

## 1. Introduction

Clear aligner therapy is an orthodontic treatment method that has been experiencing increased demand since it was introduced by Kesling in 1945 and then adapted to and incorporated into modern technologies by Align Technology in the late 1990s [1,2]. It involves the use of a series of clear, personalized, removable thermoplastic tooth positioners to progressively move misaligned teeth [1,2]. It is employed for mild orthodontic tooth movements and represents a comfortable and efficient treatment alternative to conventional orthodontic appliances [1,2]. Because of their improved esthetics and oral hygiene, the use of aligners has encouraged people of all ages, including older adults, to seek treatment for their malocclusion [3].

Clear aligners are produced via a thermoforming process at approximately 220 °C with varying pressure and holding times depending on the thermoforming foils. Currently, thermoplastic materials used for aligners include polyethylene terephthalate (PET-G), polypropylene, polycarbonate (PC), thermoplastic polyurethanes (TPU), and ethylene vinyl acetate and are also available in different layer thicknesses.

For the fabrication of aligners, dental models including a digital and a physical model are required. The digital model is essential for virtual, software-based treatment planning, whereas the physical model is essential for fabricating the orthodontic appliances via thermoforming. Plaster models fabricated from conventional impressions are considered the gold standard of physical models and can be digitized using a dental desktop scanner to create 3-dimensional (3D) digital models [4]. However, the fabrication of plaster models is time-consuming, labor-intensive, and may incorporate many errors resulting from the consecutive clinical laboratory procedures needed [5,6]. Furthermore, plaster models are heavy, bulky, require storage space, and are liable to damage and difficult to share with other professionals involved in the dental care of patients [6,7].

Intraoral scanning of the dentition is a direct method of digital model acquisition that can be used to avoid the disadvantages associated with plaster models. Evidence is growing that the intraoral scanning method is accurate and that digital dental models from intraoral scans can replace plaster models [8]. Three-dimensional printing can be used to transfer digital models obtained by intraoral scanning into physical models through a layer-by-layer deposition process. By using this process, several steps of the traditional manufacturing process for dental models can be omitted, and physical models can be fabricated in an easy and cost-effective way.

Three-dimensional printing, also known as additive manufacturing or rapid prototyping, has been applied in dentistry for more than 20 years [9]. Its application ranges from prosthodontics, oral and maxillofacial surgery, and oral implantology to orthodontics, endodontics, and periodontology. The main application of dental 3D printing is in models that are largely produced using stereolithography (SLA) or digital light processing (DLP) technologies [9,10]. Both technologies are based on light-curing polymers that are layered to a 3D object using a light source. Alternatively, models can also be additively manufactured using a fused filament fabrication (FFF). This process is based on a thermoplastic polymer filament that is extruded and built up layer-by-layer to form a 3D object. The FFF process is characterized by lower acquisition costs as well as lower costs for consumables. However, for a long time, the process was considered too inaccurate to produce dental precision models [9].

Few studies have validated the accuracy of 3D-printed models, some comparing different 3D printing technologies [11,12] and others comparing the accuracy of 3D-printed models to plaster models [6,7,13]. The presented results are conflicting. One study compared the precision and trueness of dental models manufactured with either SLA, DLP, FFF, or PolyJet technology and found differences in precision and trueness between the varying 3D printing technologies, with PolyJet and DLP technologies being more precise than FFF and SLA techniques, and with PolyJet technology having the highest accuracy [11]. Others found DLP technology more accurate than SLA and PolyJet but stated that all 3D-printed models resulted in clinically acceptable accuracy [12]. Another study compared the accuracy of SLA-printed digital models to plaster models and concluded that digital models printed using SLA technology cannot replace conventional plaster models of alginate impressions in orthodontics for diagnosis and treatment planning because of their clinically relevant transversal contraction [7]. However, other studies demonstrated that 3D-printed models (using either DLP or SLA technology) could be ideal representatives when used as diagnostic tools and in medical records because of their improved or comparable accuracy compared to plaster models [6,13]. In accordance with this, two recent reviews summarized that all different types of 3D printing technologies (SLA, DLP, and FFF) can produce clinically acceptable results for orthodontic purposes [10,14], with one review stating that models deemed clinically acceptable for orthodontic purposes may not necessarily be acceptable for applications requiring high accuracy [10] and the other review asking for more studies clarifying the accuracy and added value of 3D printing technology in orthodontics [14].

A recently published review of the current state and future possibilities of the direct 3D printing of clear orthodontic aligners stated that 3D printing, in principle, is a suitable method of fabrication for clear aligners that offers several advantages over the conventional thermoforming process (e.g., higher precision, considering that there were no errors introduced during the printing of a 3D model and thermoforming). However, the limiting factor was that there are currently no approved materials for the additive manufacturing of the indicated clear aligner [3]. Thus, for the time being, the production of aligners remains a combination of additive manufacturing of models and the conventional production of aligners by means of thermoforming. In this context, it should be considered that during thermoforming, especially when heating and melting the thermoforming sheet, temperatures of up to 220 °C are reached in the heat center of the thermoforming unit. The heat center of the thermoforming unit is only a few centimeters (approximately 10 cm) away from the model (tooth arch) over which thermoforming is to be performed. This poses the problem for additively manufactured, polymer-based models that they could deform when exposed to high temperatures, as is common in the thermoforming unit or the melted thermoforming sheet, and thus negatively influences the accuracy of fit of the aligner.

For this reason, the aim of the present investigation is to analyze the impact of different FFF filaments, varying printing parameters (number of loops), different thermoforming foils, repeated thermoforming cycles, and type of jaw on the dimensional stability of FFF models for aligners and to compare their dimensional stability with that of conventional plaster models after repeated thermoforming cycles. The first hypothesis tested was that none of the parameters mentioned above would show an impact on the dimensional stability of FFF models and, further, that there would be no differences in dimensional stability when compared to conventional plaster models. In addition, the initial sets of FFF models fabricated using two different filaments (default, four loops) were compared with those of conventional plaster models. The corresponding hypothesis was that there would be no differences in precision between the different models tested.

## 2. Materials and Methods

For this in-vitro study, dental impressions were made of the maxillary and mandibular of a young female patient (Alginoplast, Kulzer, Hanau, Germany), and master plaster models (pico rock 280, Picodent, Wipperfürth, Germany) were fabricated. The maxilla was fully dentate (14 teeth in total) and symmetrical and showed anatomical occlusal surfaces. The mandible had a gap caused by a missing tooth (35, premolar) and the front was slightly interlocked (13 teeth in total). The two master models were scanned with the Ceramill Map 400 (Amann Girrbach, Koblach, Austria), and the data were converted into the STL format and saved as master STL datasets. Dental models were generated based on master STL datasets using Meshmixer (V3.5, Autodesk, San Rafael, CA, USA) and saved as STL files. With the help of the SIMPLEX sliceware (Renfert, Hitzingen, Germany), the STL data of the dental models were converted into the g-code format. These provided specific printing instructions to the FFF printer.

### 2.1. Fabrication of Models

In total, N = 96 FFF models (maxilla and mandible) were fabricated from two different FFF filaments and four varying printing parameters (number of loops), as well as N = 12 (maxilla and mandible) conventional plaster models (Figure 1, Table 1).

For each FFF filament, namely the SIMPLEX aligner model (Renfert, Hilzingen, Germany) and Renfert PLA HT (Renfert, Hilzingen, Germany), n = 48 models were printed with a filament diameter of 1.75 mm using the printer SIMPLEX (Renfert, Hilzingen, Germany). The technical specifications of the printer are shown in Table 2. In this 3D printer, the extruder moves along the X- and Y-axes while the building platform moves along the Z-axis. Before the printing process, the filaments were not dried, and printing parameters were selected uniformly for both filaments. From the default printing parameters, only the loop count was altered, reducing one loop per group at a time. After the printing process, all models were carefully and immediately removed from the building platform by hand and cooled down to room temperature.

For the control group, the master models were duplicated using silicone molds (Adisil rose, Siladent, Goslar, Germany) that were filled with plaster (pico rock 280, Picodent, Wipperfürth, Germany). In summary, 12 plaster models were fabricated.

### 2.2. Thermoforming of the Models

All models were thermoformed (pressure molded) three times in total using two different thermoforming foils. Half of the models were thermoformed with 0.75 mm × 125 mm Ø aligner foil (CA Pro+ Clear Aligner, Scheu Dental, Iserlohn, Germany) and the other half with a 1.0 mm × 125 mm Ø Duran foil (Duran+, Scheu Dental). To guarantee the same initial height of the models, they were positioned on a metal plate associated with the thermoforming unit (BIOSTAR VII, Scheu Dental). The foil was heated in the thermoforming unit according to the manufacturer’s instructions and then placed over the models and put under air pressure (5.6 bar). The aligner foil was heated at a temperature of 220 °C for 25 s and the Duran foil at 220 °C for 30 s. The cooling time was 60 s at a pressure of 5.6 bar for both thermoforming foils. After thermoforming, all foils were carefully removed from the models using a scalpel (Figure 2).

### 2.3. Digitization of Models

All models were scanned (Ceramill Map 400) directly after fabrication as well as after each thermoforming process. In total, each model was scanned four times. Therefore, the models were positioned on spacer plates 1 and 2 as well as fixed with modeling clay to reach the height of the scan area. The scanned area was then defined in the scanning software (Ceramill Map, Amann Girrbach). The scan was carried out all around each single tooth with a defined high scan quality (repeat accuracy of 6 µm). The obtained STL datasets were imported into a 3D data analysis software for superimposition.

### 2.4. 3D Data Analysis Using Step Models

Analysis of the dimensional stability of the different models after thermoforming as a function of the thermoforming foil used was performed using a step model in GOM Inspect Pro 2021 (Carl Zeiss GOM Metrology, Braunschweig, Germany). A step model was created for each printed model.

For this, the initial situation (pre-scan) was first imported as a mesh. If present, mesh errors were corrected, and subsequently, the mesh was converted into CAD as a reference model. Then, the dataset of the model after the first thermoforming cycle was imported as a mesh, and again, mesh errors were corrected if present. Then, the two imported datasets of the models were aligned in two steps. First, the automatic pre-alignment was applied, in which the mesh was pre-aligned to the CAD. The searching time was set to “normal” in the settings. In the next step, the main alignment was performed using the local best-fit method. For this purpose, the model base and the gingival areas were selected as the alignment area, and the initial situation (CAD) was defined as the target element (maximum distance was set to 1 mm). The teeth were defined as the inspection area. For the inspection, the area previously defined for alignment was restored under “local best fit” and was inverted so that the area of interest for the analysis of dimensional stability was selected. To analyze deviations between CAD and mesh, an area comparison was applied to the mesh (maximum distance was set to 10 mm). Selected surfaces that were used for local best fit and surface inspection, as well as visualization of results, are shown to be exemplary for the mandibular model in Figure 3. To visualize deviations, the minimum and maximum were displayed in color and localized using deviation flags. Unrealistically high or low deviation flags or extreme changes in the color gradient suggested errors in the mesh. If this was the case, the mesh errors were corrected via repairing, smoothing, or refining. Then, the step model was generated by importing the dataset of the model after the second thermoforming cycle as a mesh to generate the “second step” in the analysis software. For this dataset, mesh errors were corrected if present. The alignment and selection of the inspection area was automatically set in the step model as equal to the “first step”. This procedure was repeated for another dataset of the model, namely after the third thermoforming cycle as the “third step”. In every single step, the deviations of each model after thermoforming (mesh) from the initial model (CAD) were analyzed. Finally, the maximum negative deviations per analysis step were exported for each test group and transferred to a coded Excel sheet.

### 2.5. Statisical Analysis

Data were descriptively analyzed, and the Kolmogorov–Smirnov test was computed to test for a violation of the normal distribution. The effect of different variables on dimensional stability was examined using a global univariate analysis with a post-hoc Scheffé test, testing the effect size (η_p_^2^). Data were split, and significant differences between groups were determined using a one-way ANOVA, post-hoc Scheffé test, and *t*-test on independent samples. For these tests, *p* values below 0.05 were regarded as statistically significant (IBM Statistics SPSS 27.0, IBM, Amonk, NY, USA). A descriptive comparison between the FFF and plaster models using 95% confidence intervals was performed.

## 3. Results

In summary, 14 groups (out of 111) showed a large deformation of the models (SIMPLEX aligner model filament: 3× Renfert PLA HT: 11×) after thermoforming and could not be evaluated for deviations because superimposition of the data failed (Table 3). These groups were not included in the statistical analysis. Approximately 10% (11/97) of the tested groups showed deviations from the normal distribution. Data was analyzed parametrically.

The dimensional stability of the models was most strongly affected by the printing parameters (number of loops; η_p_^2^ = 0.768, *p* < 0.001) followed by the thermoforming foil used (η_p_^2^ = 0.663, *p* < 0.001) as well as the type of model (η_p_^2^ = 0.588, *p* < 0.001). In addition, various interactions showed an influence on dimensional stability (η_p_^2^ = 0.041–0.386, *p* < 0.035). Therefore, the dataset was split for further analysis.

### 3.1. Impact of Thermoforming Foil

Within the SIMPLEX aligner model filament, one loop in maxillary models (*p* = 0.020–0.034) and two loops (*p* = 0.015–0.041) in mandibular models showed lower dimensional stability within groups thermoformed using Duran foil than within those thermoformed using aligner foil, regardless of the thermoforming cycles. The same differences were observed after the third thermocycling cycle within the mandibular models and default group (*p* = 0.045). Within the Renfert PLA HT filament, thermoforming with the Duran foil showed generally lower dimensional stability compared to thermoforming using aligner foil (*p* < 0.001–0.043). No impact of thermoforming foil was observed after the first thermoforming cycle with the default maxillary group, after the second thermoforming cycle for the mandibular-fabricated group with two loops, or after the third thermoforming cycle for maxillary default models and mandibular models printed with three loops (*p* > 0.05). Within plaster models, groups thermoformed using aligner foil showed higher dimensional stability after the second (*p* = 0.040) and third (*p* = 0.013) thermoforming cycles compared to models thermoformed with Duran foil.

The impact of thermoforming foil is shown in Figure 4 as a comparison of one aligner foil versus Duran foil. This example shows the effects after the third thermoforming cycle using the SIMPLEX aligner model filament and one loop for printing. The greatest deformation resulted from using the Duran foil and was located in vestibular and occlusal areas.

### 3.2. Impact of FFF Filament

For maxillary and mandibular models printed with one loop or two loops, the SIMPLEX aligner model showed higher dimensional stability compared to Renfert PLA HT (*p* < 0.008) regardless of the thermoforming foil and the thermoforming cycles. In contrast, in mandibular models printed with one loop and thermoformed with aligner foil, no impact of the FFF filament was observed within the first or third thermoforming cycle (*p* > 0.125). Additionally, no impact of the FFF filament was observed in maxillary and mandibular models printed with two loops within first thermoforming cycle (*p* > 0.292). For mandibular models printed with three loops and thermoformed using aligner foil, an impact of the FFF filament was determined (*p* < 0.022) with the SIMPLEX aligner model resulting in higher dimensional stability compared to Renfert PLA HT, irrespective of the thermoforming cycles. In turn, no differences were observed between maxillary models (*p* > 0.165), regardless of the thermoforming cycle. Within all mandibular models printed with four loops, the SIMPLEX aligner model showed higher dimensional stability compared to Renfert PLA HT, regardless of the thermoforming foil (*p* < 0.049). The same results were observed within maxillary models printed with four loops after the third thermoforming cycle using aligner foil (*p* = 0.018) and after the second thermoforming cycle using Duran foil (*p* < 0.001). The remaining groups presented no impacts of FFF filament on their dimensional stability (*p* > 0.053). However, especially for models fabricated from Renfert PLA HT and printed with one loop, the deformation observed was so high that the datasets could not be superimposed and, accordingly, could not be evaluated. The impact of the filament is shown in Figure 5 as a comparison of the SIMPLEX aligner model filament versus the Renfert PLA HT filament. 

This example shows the effects after the third thermoforming cycle using aligner foil for thermoforming and one loop for printing. The greatest deformation resulted from using the Renfert PLA HT filament and was located all over the dental arch, particularly in the vestibular on 33 and 43.

### 3.3. Impact of Printing Parameter (Number of Loops)

In general, the increase in the number of loops increased the deformation stability, regardless of the model jaw. One loop showed lower deformation stability than did three loops within maxillary models fabricated from the SIMPLEX aligner model and thermoformed with aligner foil (*p* < 0.001–0.022). Within groups thermoformed using Duran foil, two loops or more showed higher deformation stability than one loop (*p* < 0.001–0.019). No impact of the loops was observed for the SIMPLEX aligner mandibular models thermoformed using aligner foil within the first thermoforming cycle (*p* = 0.140) or for SIMPLEX aligner maxillary models thermoformed using Duran foil within the third thermoforming cycle (*p* = 0.142).

Within models fabricated from Renfert PLA HT, printing with one loop led to lower deformation stability than printing with two, three, or four loops (*p* < 0.001–0.022). No impact of the printing parameters (number of loops) was observed for mandibular models fabricated from Renfert PLA HT and thermoformed using Duran foil within the second thermoforming cycle (*p* = 0.096), as well as for maxillary models of the same group within the first (*p* = 0.084) and third thermoforming cycles (*p* = 0.236).

The impact of the number of loops is shown in Figure 6 as a comparison of one loop versus four loops. This example shows the effects after the third thermoforming cycle using Duran foil for thermoforming and the SIMPLEX aligner model filament for printing. The greatest deformation resulted from printing with one loop and was located in the vestibular and occlusal areas.

### 3.4. Impact of Thermoforming Cycles

Within the SIMPLEX aligner model filament, no statistical impact of thermoforming cycles was found. However, maxillary models printed with the default parameter (four loops) and thermoformed using Duran foil could not be evaluated after the third thermoforming cycle. The same applied to mandibular models printed with one loop, regardless of the thermoforming cycle. Within the Renfert PLA HT filament, the default and two-loop maxillary and mandibular models thermoformed using aligner and the mandibular default models showed a decrease in deformation stability with an increase in the thermoforming cycles (*p* < 0.049). Within this FFF filament, numerous groups could not be evaluated (Table 3).

Within the plaster models, no impact of thermoforming cycles was determined (*p* = 0.251–0.996).

The impact of the thermoforming cycle is shown in Figure 7 as a comparison of the first cycle and third cycle. This example shows the effect of using Duran foil for thermoforming, the Simplex filament, and one loop for printing. The greatest deformation resulted from the third thermoforming cycle and was located in the vestibular and occlusal areas.

### 3.5. Impact of Model Jaw

Within the SIMPLEX aligner models thermoformed using aligner foil, maxillary models showed a higher deformation stability than did mandibular models after the first (*p* = 0.019) and third (*p* = 0.025) thermoforming cycles. For all Renfert PLA HT models printed with default parameters (four loops) and thermoformed using aligner foil (*p* < 0.001), as well as for all plaster models after the second thermoforming cycle (*p* = 0.038), maxillary models showed a lower deformation stability than did mandibular models. In contrast, within Renfert PLA HT models printed with default parameters (four loops) and thermoformed using Duran foil, mandibular models showed a lower deformation stability than maxillary models (*p* = 0.010) after the second thermoforming cycle.

The impact of the jaw is shown in Figure 8 as a comparison of models of mandibula versus maxilla. This example shows the effect of using aligner foil for thermoforming, the Simplex filament, and four loops for printing. The greatest deformation was found in the maxillary model and was located on incisor edges.

### 3.6. Comparison with Plaster Models

When pooling all plaster models, a mean deviation of −0.2975 mm with a 95% confidence interval [−0.40;−0.19] was calculated. Only the SIMPLEX aligner models (four-loop default) thermoformed on maxillary models using aligner foil showed higher deformation stability than did the plaster models. In the same range of values were the following groups:All SIMPLEX aligner model models with the printing parameter of two, three, or four (default) loops and thermoformed on maxillary and mandibular (except four-loop models) models using aligner foil;All four-loop and sometimes three-loop SIMPLEX aligner models thermoformed on maxillary and mandibular models using Duran foil;All four-loop (default) maxillary models up to thermoforming cycle two and all four- (default) and three-loop Renfert PLA HT filament models thermoformed on mandibular models using aligner foil.

## 4. Discussion

The aim of the present investigation was to test the impact of FFF filaments, printing parameters, thermoforming foils, repeated thermoforming cycles, and type of jaw on the dimensional stability of FFF models for aligners and to compare their dimensional stability with that of conventional plaster models after repeated thermoforming.

Based on the results found in the present investigation, both null hypotheses—being firstly that none of the parameters tested would have an impact on dimensional stability and secondly that there would be no differences in dimensional stability between FFF models and plaster models for aligners—must be rejected.

### 4.1. Impact of Model Type and FFF Filament

Principally, the overall results of the present investigation are trivial, comprehensible, and logical. First, no differences between the initial sets of plaster models and the different FFF models, which were fabricated with the printing parameters defined by the manufacturer (default parameters of loop counts), were observed.

However, it was found that the dimensional stability of FFF models decreases with an increasing number of thermoforming cycles. This observation can be attributed to the fact that the printing temperature of the filaments (nozzle temperature of the SIMPLEX aligner model: 240 °C; Renfert PLA HT: 225 °C) and the temperature for melting the thermoforming foils (220 °C) are very close to each other. Because the thermoforming foils are heated for only a few seconds (25–30 s depending on the thickness of the thermoforming foil), the influence is correspondingly low for one-time thermoforming, but repeated thermoforming seems to have an increasing impact on the dimensional stability of the models, especially with the Renfert PLA HT filament. The fact that models made of Renfert PLA HT filament showed lower dimensional stability when exposed to repeated thermoforming could be due to differences in the materials of the filaments, considering that the diameters of the filaments were identical (1.75 mm). Renfert PLA HT consists of polylactide (PLA). PLA in general has a glass transition temperature of 60 °C and a melting interval of 175–220 °C [15]. Because the maximum temperature of the melting interval corresponds to the melting temperature of the thermoforming foils, the impact on the dimensional stability of Renfert PLA HT models is obvious. In contrast, the SIMPLEX aligner model filament consists of glycol-modified polyethylene terephthalate PETG, which is one of the most used thermoplastics in 3D printing as a result of its high impact resistance and ductility.

PETG has a glass transition temperature of approximately 80–85 °C and a melting interval of 235–255 °C. Because the melting temperature is further from the heating temperature of the thermoforming sheets, a weaker influence on the dimensional stability of the SIMPLEX aligner model models is likely, even after repeated thermoforming.

A recent investigation analyzed the dynamic mechanical and thermal properties of PETG aligners after thermoforming and artificial aging and demonstrated that thermoforming had a more prominent role than aging in diminishing the thermomechanical properties of thermoplastic polymer [16]. Both the flexural modulus and hardness revealed a significant decrease from thermoforming just as they had from the dynamic glass transition temperature. The latter observation was said to indicate the attenuation of thermomechanical properties after this process. In general, it was noted that some recent scientific investigations have focused on the impact of temperature on the printability and the related material properties of PETG [17,18,19]. Guessasma et al. determined a glass transition temperature of 77 °C for PETG, which is consistent with the information on the material in the present study and indicates that it can only be printed within a limited range of printing temperatures above 230 °C [17]. Furthermore, they found that FFF processing of PETG resulted in a significant reduction in elongation at break and a significant loss in both stiffness (up to 40%) and tensile strength (up to 40%) of PETG. At a printing temperature of 250 °C, a slight improvement in the tensile strength of PETG was observed, but the loss in both toughness and tensile strength still accounts for one-third of the raw material [17]. These findings are in general agreement with the research results of Valvez et al. [19]. Here, the optimum nozzle temperature was identified as slightly higher at 265 °C, with further dependence on the printing speed and the layer thickness, to achieve the highest possible bending properties for neat PETG. However, again, the study outcomes indicate that temperature is one of the most relevant parameters in determining the mechanical properties of PETG when processed using FFF technology. Furthermore, the research group of Hsueh et al. also investigated the influence of printing parameters (temperature and speed) on the thermal and mechanical properties of PETG and PLA using FFF technology [18]. Both polymers are also subjects of the present investigation. Hsueh et al. showed that the mechanical properties of PLA and PETG increased with increasing printing temperatures. The same observation was found for PLA with increasing printing speed, whereas for PETG, a decrease in mechanical properties was identified as printing speed increased. However, in relation to the present study, the most interesting finding was that PLA has higher mechanical properties compared to PETG but is less resistant to thermal deformation [18], which is in perfect agreement with the observations of the present investigation. In general, the findings from the existing literature, that pressure parameters have a significant impact on the mechanical and thermal properties of FFF filaments, are consistent with the observations of the present investigation. Here, too, it has been shown that it is not the FFF filament itself but the related printing parameters (loop counts) that have the greatest influence on dimensional stability.

Overall, it would be reasonable to verify the present observations via polymer analytics using FTIR or GPC analysis, as well as by examining further filaments from other manufacturers intended for the fabrication of models for aligners. Both the limited information available from the manufacturer for the investigated polymers and the fact that the present investigation examined only two filaments from one and the same manufacturer must be considered a limitation of the study. Both limitations should be considered in future studies. Furthermore, future study designs could benefit from considering alternative, already established additive manufacturing methods of models such Vat lithography-based AM technology and polymer jetting-based AM technology.

### 4.2. Impact of Printing Parameters (Loop Counts)

In the present investigation, printing parameters (loop counts) were reduced from the default parameters specified by the manufacturer to determine the extent to which the loop count can be reduced in order to always ensure sufficient dimensional stability of FFF models for the fabrication of aligners. The loop count represents the number of extruded lines (perimeter + loops) that the vertical shell is made of and defines the shell thickness on the side of the object. The outermost vertical shell is the perimeter, and the remaining optional inner vertical shells are called loops. A reduction in the loop count has economic advantages, as both printing time and filament are saved. Table 4 shows a comparison of the printing time and the amount of filament used depending on the jaw and the loop count. The percentage of time and material savings in relation to the default parameters is between 4 and 7% and 7 and 8%, respectively, regardless of the filament used. A reduction in loop counts is accompanied by a lower mass of the FFF models, which in turn makes the model more susceptible to deformation when exposed to certain temperatures. This fact is also reflected in the present results on the dimensional stability of maxillary and mandibular models. Although no differences in dimensional stability were identified between maxillary and mandibular models, it was nevertheless observed that mandibular models, which are less massive compared to maxillary models, tend to deform more. 

Regarding the reduction of printing parameters (loop counts), it was found that models made of the Renfert PLA HT filament, in particular, deform more strongly. To ensure sufficient dimensional stability even after repeated thermoforming, it is recommended not to reduce the printing parameters (loop counts) to less than three loops. The reduction in the printing parameters to three loops corresponds to a time saving of 4–7% and a material saving of 7–8%, depending on the model’s jaw, compared to the default parameters (four loops).

### 4.3. Impact of Thermoforming Foil

Last but not least, the dimensional stability of the FFF models was affected by the thermoforming foil. Here, a greater impact on the dimensional stability was observed with an increasing thickness of the thermoforming foil. This effect was especially observed in the less dimensionally stable filament Renfert PLA HT irrespective of the loop counts, as well as for mandibular models made from the SIMPLEX aligner model in the considerably reduced loop counts of one and two. As already mentioned, both foils melted at an identical temperature (220 °C). There was a difference of 5 s in the heating time, which appears proportional with respect to the higher foil thickness of 0.25 mm. The pressure of 5.6 bar during cooling was also identical for both thermoforming foils. For this reason, the lower dimensional stability of the models thermoformed with a thicker thermoforming sheet can be attributed solely to the layer thickness. It is conceivable that the foil with the higher layer thickness has a higher heat capacity and transmits more temperature to the model being thermoformed. Further, because of the higher layer thickness, greater stress is generated during cooling under the pressure of 5.6 bar, which favors the deformation of the FFF models, especially when the melting interval of the FFF filament is in a comparable temperature range with the melting temperature of the thermoforming foil.

Even though FFF models seem to be more susceptible to deformation under repeated exposure to temperature compared to plaster models, plaster models are more prone to broken anterior teeth, making the models unusable. For this reason, plaster models are often duplicated, which requires a lot of work. From this point of view, FFF models are characterized by their convenient, cost effective, and reproducible fabrication, which is most beneficial to the patient as no further impression taking is required.

## 5. Conclusions

Within the limitations of this investigation, the following conclusions can be drawn:The initial set of FFF models are comparable to the gold standard “plaster models”, but the dimensional stability of FFF models decreases with increasing numbers of thermoforming cycles.Both the filament used and the loop count have a significant impact on dimensional stability, with Renfert PLA HT being less stable than the SIMPLEX aligner model filament, and with loop counts of three and four showing comparable dimensional stability for the Simplex aligner.The higher the layer thickness of the thermoforming foil, the greater the impact on the dimensional stability of FFF models.

## Figures and Tables

**Figure 1 materials-16-04835-f001:**
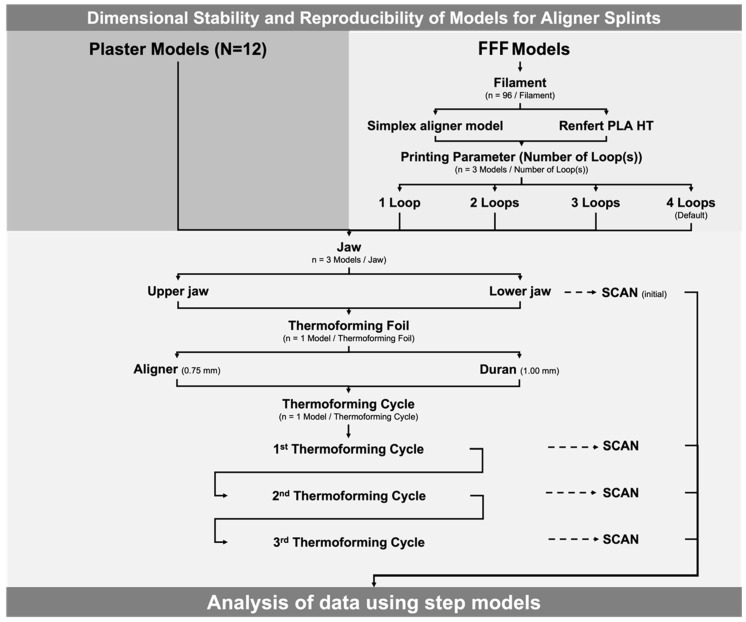
Study design.

**Figure 2 materials-16-04835-f002:**
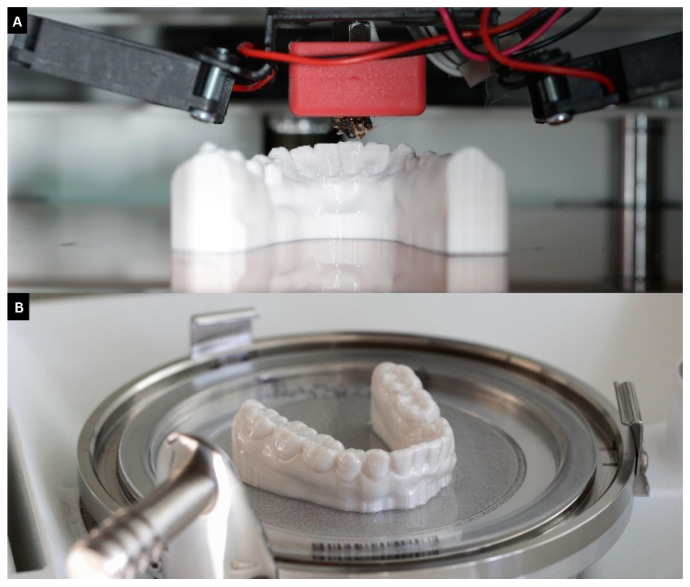
Printed FFF model on printing plate (**A**), model with thermoforming foil (**B**).

**Figure 3 materials-16-04835-f003:**
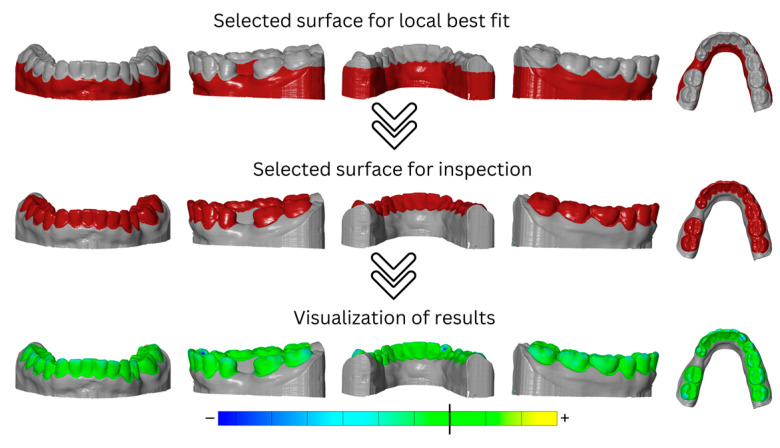
Areas used for surface selection for local best fit (**top**, red), surface inspection (**center**, red) and visualization of results as heatmap (**bottom**) are shown to be exemplary for mandibular model.

**Figure 4 materials-16-04835-f004:**
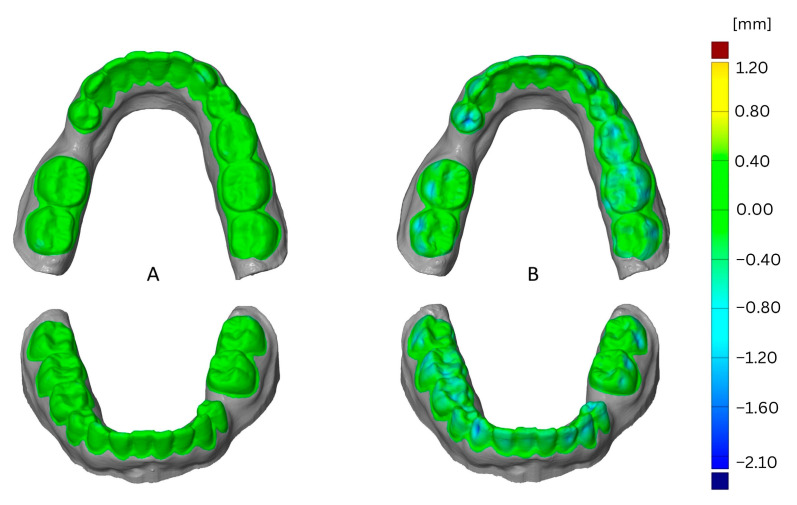
Comparison of deformation resulting from thermoforming foil, CA^®^ Pro+ (Aliner, (**A**)), versus DURAN^®^ (**B**).

**Figure 5 materials-16-04835-f005:**
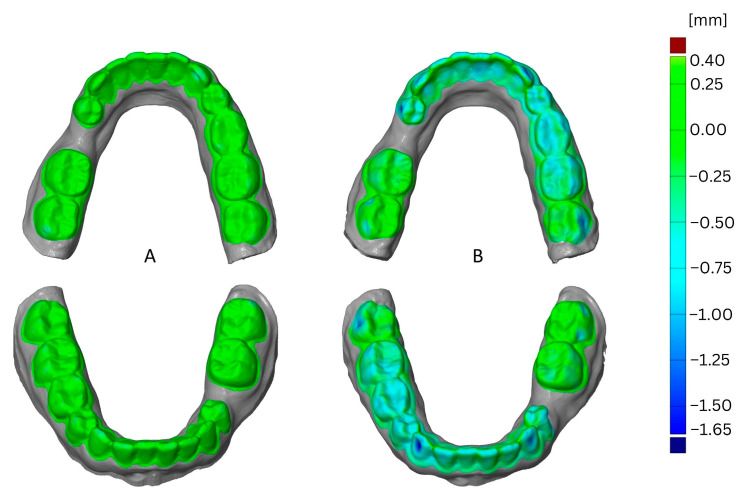
Comparison of deformation resulting from filaments, SIMPLEX aligner model (**A**) versus Renfert PLA HT (**B**).

**Figure 6 materials-16-04835-f006:**
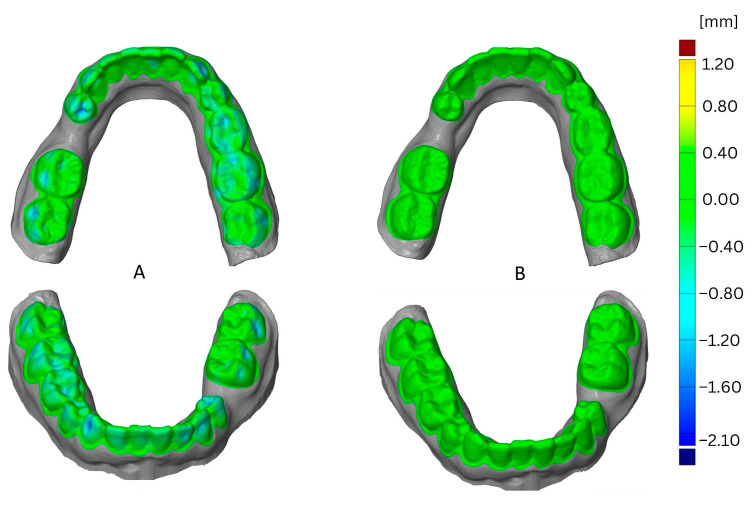
Comparison of deformation resulting from loops, one loop (**A**) versus four loops (**B**).

**Figure 7 materials-16-04835-f007:**
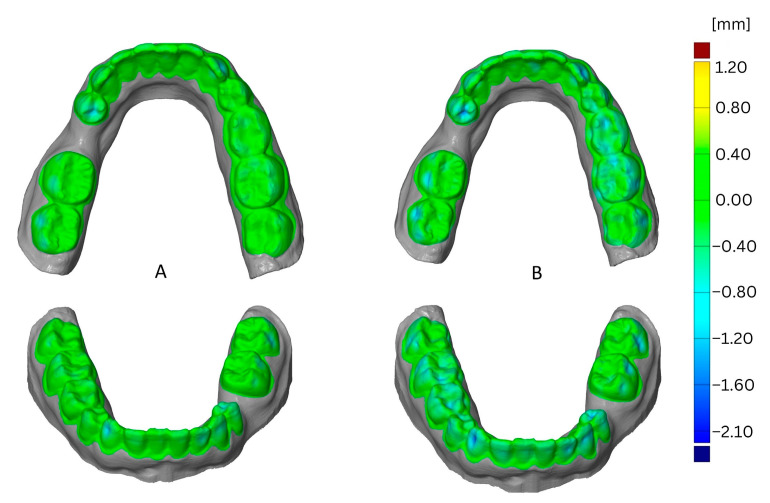
Comparison of deformation resulting from thermoforming cycle, first (**A**) versus third (**B**).

**Figure 8 materials-16-04835-f008:**
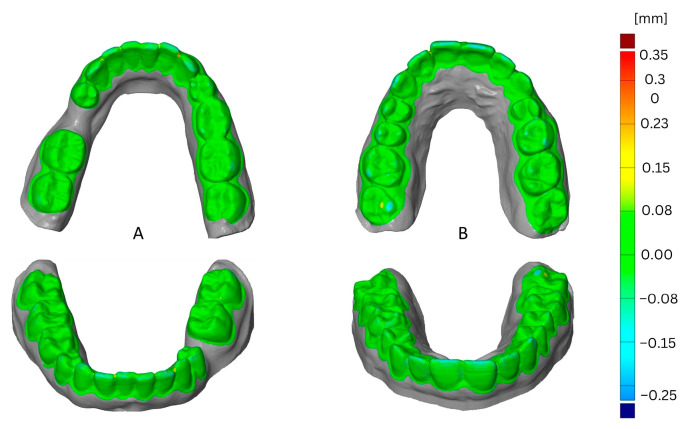
Comparison of deformation resulting from jaw, mandibula (**A**) versus maxillae (**B**).

**Table 1 materials-16-04835-t001:** List of used materials sorted by type of material, product name, manufacturer, and LOT number.

Type of Material	Product Name	Material	Manufacturer	LOT
FFF Filament	Simplex aligner model	PETG	Renfert, Hilzingen, Germany	17350300
	Renfert PLA HT	PLA	Renfert, Hilzingen, Germany	EP90213−1 L
Plaster	Pico-rock 280	N/A	Picodent, Wipperfürth, Germany	220112507
Thermoforming Sheet	CA^®^ Pro+ Aligner (0.75 mm)	N/A	Scheu Dental, Iserlohn, Germany	0522A
	DURAN^®^+ (1.00 mm)	N/A	Scheu Dental, Iserlohn, Germany	1022A

**Table 2 materials-16-04835-t002:** Technical specifications of the 3D printer.

Size	250 × 200 × 200mm
Layer resolution	≥50 microns
Positioning precision (x, y, z)	4 × 4 × 2 microns
Filament diameter	1.75 mm
Printing speed	50–200 mm/s
Type of extruder	Single, all-metal hot end
Nozzle operating temperature	180–260 °C
Nozzle diameter	0.4 mm
Ambient temperature	15–32 °C
Heated building platform	50–110 °C
Ventilation	Yes
Slicing software	SIMPLEX sliceware

**Table 3 materials-16-04835-t003:** Descriptive statistics showing the mean deviation (mm) after each thermoforming cycle depending on the model type, thermoforming sheet, model jaw, and printing parameters (if applicable).

Model Type	Thermoforming Foil	Model Jaw	Printing Parameters	Deviation in mm (Mean ± SD)/Relative Frequency (95% CI)		
				Thermoforming Cycle 1	Thermoforming Cycle 2	Thermoforming Cycle 3
SIMPLEX aligner model	CA^®^ Pro+ Aligner	OK	Default (4 Loops)	−0.300 ± 0.052 * ^a;AB;i;I;1^/100 (29;100)	−0.280 ± 0.079 ^a;B;i;I;1,2^/100 (29;100)	−0.297 ± 0.067 ^b;B;I;2^/100 (29;100)
			3 Loops	−0.217 ± 0.051 ^b;B;i;I^/100 (29;100)	−0.233 ± 0.075 ^b;B;i;I^/100 (29;100)	−0.257 ± 0.059 ^b;B;i;I^/100 (29;100)
			2 Loops	−0.253 ± 0.051 ^b;AB;i;I^/100 (29;100)	−0.277 ± 0.074 ^b;B;i;I^/100 (29;100)	−0.277 ± 0.055 ^b;B;i;I^/100 (29;100)
			1 Loop	−0.483 ± 0.178 ^b;A;ii;I^/100 (29;100)	−0.543 ± 0.151 ^b;A;ii;I^/100 (29;100)	−0.570 ± 0.161 ^b;A;ii;I^/100 (29;100)
		UK	Default (4 Loops)	−0.153 ± 0.042 ^b;A;i;I;†^/100 (29;100)	−0.157 ± 0.038 ^b;B;i;I;2^/100 (29;100)	−0.157 ± 0.021 ^b;B;ii;I;2^/100 (29;100)
			3 Loops	−0.290 ± 0.040 ^a;A;i;I^/100 (29;100)	−0.300 ± 0.010 ^a;AB;i;I^/100 (29;100)	−0.313 ± 0.006 * ^a;AB;i;I^/100 (29;100)
			2 Loops	−0.237 ± 0.093 ^b;A;ii;I^/100 (29;100)	−0.257 ± 0.074 ^b;B;ii;I^/100 (29;100)	−0.317 ± 0.081 ^b;AB;ii;I^/100 (29;100)
			1 Loop	−0.593 ± 0.412 ^a;A;i;I^/100 (29;100)	−0.713 ± 0.308 ^a;A;i;I^/100 (29;100)	−0.720 ± 0.329 ^a;A;I^/100 (29;100)
	DURAN^®^+	OK	Default (4 Loops)	−0.283 ± 0.081 ^a;B;i;I;1^/100 (29;100)	−0.247 ± 0.051 ^b;B;i;I;1^/100 (29;100)	N/A ^a;†^/33.3 (0;91)
			3 Loops	−0.430 ± 0.161 ^b;B;i;I^/100 (29;100)	−0.383 ± 0.075 * ^b;B;i;I^/100 (29;100)	−0.393 ± 0.101 ^a;A;i;I^/100 (29;100)
			2 Loops	−0.670 ± 0.367 ^a;AB;i;I^/100 (29;100)	−0.550 ± 0.170 ^b;B;i;I^/100 (29;100)	−1.003 ± 0.794 ^b;A;i;I^/100 (29;100)
			1 Loop	−1.123 ± 0.302 ^b;A;i;I^/100 (29;100)	−1.373 ± 0.394 ^b;A;i;I^/100 (29;100)	−1.450 ± 0.370 ^b;A;i;I^/100 (29;100)
		UK	Default (4 Loops)	−0.337 ± 0.120 ^b;B;i;I;1,2^/100 (29;100)	−0.340 ± 0.115 ^b;B;i;I;2^/100 (29;100)	−0.343 ± 0.111 ^b;A;i;I;2^/100 (29;100)
			3 Loops	−0.337 ± 0.093 ^b;B;i;I^/100 (29;100)	−0.423 ± 0.145 ^b;B;i;I^/100 (29;100)	−0.453 ± 0.163 ^b;A;i;I^/100 (29;100)
			2 Loops	−0.570 ± 0.171 ^a;B;i;I^/100 (29;100)	−0.680 ± 0.165 ^a;B;i;I^/100 (29;100)	−0.707 ± 0.162 ^a;A;i;I^/100 (29;100)
			1 Loop	−1.447 ± 0.509 ^a;A;i;I^/100 (29;100)	−1.895 ± 0.629 ^b;A;i;I^/66.7 (9;100)	N/A ^b^/33.3 (0;91)
Renfert PLA HT	CA^®^ Pro+ Aligner	OK	Default (4 Loops)	−0.357 ± 0.006 * ^a;B;i;I;1^/100 (29;100)	−0.400 ± 0.026 ^a;B;ii;II;1^/100 (29;100)	−0.447 ± 0.006 * ^a;C;i;III;1^/100 (29;100)
			3 Loops	−0.427 ± 0.085 ^a;B;ii;I^ 100 (29;100)	−0.490 ± 0.040 ^a;B;ii;I^/100 (29;100)	−0.543 ± 0.031 ^a;B;I^/100 (29;100)
			2 Loops	−0.400 ± 0.010 ^a;B;ii;II^ 100 (29;100)	−0.657 ± 0.067 ^a;A;I^/100 (29;100)	−0.750 ± 0.036 ^a;A;I^/100 (29;100)
			1 Loop	−1.390 ± 0.366 ^a;A^ 100 (29;100)	N/A ^a^/33.3 (0;91)	N/A ^a^/0 (0;71)
		UK	Default (4 Loops)	−0.237 ± 0.015 ^a;B;ii;II;†^/100 (29;100)	−0.300 ± 0.026 ^a;B;ii;I;1^/100 (29;100)	−0.327 ± 0.015 ^a;A;ii;I;1^/100 (29;100)
			3 Loops	−0.283 ± 0.012 * ^a;B;ii;I^/100 (29;100)	−0.397 ± 0.098 * ^a;AB;ii;I^/100 (29;100)	−0.067 ± 0.508 * ^a;A;i;I^/100 (29;100)
			2 Loops	−0.393 ± 0.006 * ^a;AB;i;II^/100 (29;100)	−0.557 ± 0.096 ^a;A;i;I;II^/100 (29;100)	−0.617 ± 0.090 ^a;A;I^/100 (29;100)
			1 Loop	−0.877 ± 0.426 ^a;A;ii^/100 (29;100)	N/A ^a^/33.3 (0;91)	N/A ^a^/33.3 (0;91)
	DURAN^®^+	OK	Default (4 Loops)	−0.560 ± 0.156 ^a;A;i;I;1^/100 (29;100)	−0.610 ± 0.036 ^a;B;i;I;1^/100 (29;100)	−0.645 ± 0.078 * ^a;i;I;†^/66.7 (9;100)
			3 Loops	−0.863 ± 0.188 ^a;A;i;I^/100 (29;100)	−0.973 ± 0.162 ^a;A;i;I^/100 (29;100)	N/A ^a^/33.3 (0;91)
			2 Loops	−0.967 ± 0.211 ^a;A;i^/100 (29;100)	N/A ^a^/33.3 (0;91)	N/A ^a^/0 (0;71)
			1 Loop	N/A ^a^/0 (0;71)	N/A ^a^/0 (0;71)	N/A ^a^/0 (0;71)
		UK	Default (4 Loops)	−0.567 ± 0.042 ^a;B;i;II;1^/100 (29;100)	−0.825 ± 0.050 ^a;A;i;I;1^/66.7 (9;100)	−0.860 ± 0.087 ^a;B;i;I;1^/100 (29;100)
			3 Loops	−0.973 ± 0.208 ^a;B;i;I^/100 (29;100)	−1.160 ± 0.102 ^a;A;i;I^/100 (29;100)	−1.230 ± 0.115 ^a;A;i;I^/100 (29;100)
			2 Loops	−0.795 ± 0.148 ^a;B;i;I^/66.7 (9;100)	−1.330 ± 0.311 ^a;A;i;I^/66.7 (9;100)	N/A ^a^/33.3 (0;91)
			1 Loop	−2.605 ± 0.262 ^a;A;i^/66.7 (9;100)	N/A ^a^/0 (0;71)	N/A ^a^/0 (0;71)
Plaster	CA^®^ Pro+ Aligner	OK	---	−0.217 ± 0.015 ^i;I;2^/100 (29;100)	−0.203 ± 0.021 ^i;I;2^/100 (29;100)	−0.233 ± 0.065 ^i;I;2^/100 (29;100)
		UK	---	−0.147 ± 0.049 ^i;I;†^/100 (29;100)	−0.130 ± 0.036 ^ii;I;2^/100 (29;100)	−0.177 ± 0.065 ^ii;I;2^/100 (29;100)
	DURAN^®^+	OK	---	−0.463 ± 0.586 * ^i;I;1^/100 (29;100)	−0.467 ± 0.592 ^i;I;1^/100 (29;100)	−0.430 ± 0.563 ^i;I;†^/100 (29;100)
		UK	---	−0.300 ± 0.121 ^i;I;2^/100 (29;100)	−0.340 ± 0.115 ^i;I;2^/100 (29;100)	−0.463 ± 0.095 ^i;I;2^/100 (29;100)

* indicates the deviation from the normal distribution; a, b indicates significant differences between the filaments within one thermoforming foil, printing parameter, model jaw, and thermoforming cycle; A, B, C indicates significant differences between the printing parameter within one filament, thermoforming foil, model jaw, and thermoforming cycle; 1, 2 indicates significant differences between the model types (filaments printed with default parameter (four loops)) within one thermoforming foil, one model jaw, and one thermoforming cycle, † significant differences could not be calculated because of the excessive deformation of the models; i, ii indicates significant differences between the thermoforming foil within model type, model jaw, printing parameter, and thermoforming process; I, II, III indicates significant differences between the thermoforming cycles within one model type, printing parameter, model jaw and thermoforming foil.

**Table 4 materials-16-04835-t004:** Overview of printing time (hh:mm:ss) and filament used (m) depending on the model’s jaw and the printing parameters (loop count), as well as percentage of savings in printing time and filament used for reduced loop counts with reference to the default printing parameters (four loops).

	Lower Jaw		Upper Jaw	
	Printing Time in hh:mm:ss (Percentage of Savings in Printing Time)	Filament Used in m (Percentage of Savings in Filament Used)	Printing Time in hh:mm:ss (Percentage of Savings in Printing Time)	Filament Used in m (Percentage of Savings in Filament Used)
4 Loops	01:39:00	4.321	01:47:00	4.890
3 Loops	01:35:00 (4%)	3.967 (8%)	01:39:00 (7%)	4.541 (7%)
2 Loops	01:23:00 (16%)	3.579 (17%)	01:31:00 (15%)	4.166 (15%)
1 Loop	01:16:00 (23%)	3.152 (27%)	01:25:00 (21%)	3.757 (23%)

## Data Availability

On reasonable request, the corresponding author will make the datasets generated during the current investigation available.

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
