# Peer review of "Dimensional Stability and Reproducibility of Varying FFF Models for Aligners in Comparison to Plaster Models"

_materials, 2023, doi:10.3390/ma16134835_

Round 1
Reviewer 1 Report
The study is well structured and understandable and very practical. It is important to investigate the fabrication of aligners using new methods to generate the best results for patients.
In the introduction it is mentioned that the models are manufactured by lithography based AM technologies (DLP, SLA, jetting) nowadays. (page 2 76-77) and that there is a great uncertainty between published data. It seems that there is no clear statement regarding the accuracies of dental models in the literature or very controversial data has been published.
In the literature in the introduction, there are no references to the stability of models fabricated using lithography-based AM technologies after thermoforming as in the submitted study. To ensure comparability with the most commonly used technologies for the fabrication of models for aligner fabrication, at least one Vat lithography-based AM technology and polymer jetting based AM technology should be included in the recent study design. Additive manufacturing of models for aligner fabrication is already standard today and must be considered when new materials and technologies are investigated as alternatives.
Why were only filaments from Renfert used? There are other filaments on the market that are designed for this indication (Filadental; Weithas; Dental 3D Agency).
I recommend making up the required groups and then resubmitting to Materials.
While reading, I noticed terms that should be checked again, such as:
· Fused Filament Fabrication (page 2 80); when checking, the "ISO/ASTM 52900 - Terminology for AM - General Principles - Terminology" should be taken into account and correct terms should be used.
· The term "Fused Deposition Modeling" is not spelled correctly (page 2 80) and is a trademark term of Stratasys and should therefore be avoided in scientific publications. Please change throughout the manuscript.
Furthermore, could the descriptive statistic in Table 3 be formatted as a graph? This would make it easier for the reader to visually examine the comparisons. Examples can be found in Son, K.; Lee, W.-S.; Lee, K.-B. Effect of Different Software Programs on the Accuracy of Dental Scanner Using Three-Dimensional Analysis. Int. J. Environ. Res. Public Health 2021, 18, 8449. https:// doi.org/10.3390/ijerph18168449.
Thanks for the interesting manuscript.
Minor check
Author Response
Revision of manuscript: materials-2471594
Title: Dimensional stability and reproducibility of varying FDM models for aligners in comparison to plaster models
Dear Editor,
We thank you and the reviewer for your suggestions, insightful comments and corrections. We appreciate your efforts to evaluate our manuscript and providing us with the opportunity to make the following revisions. We have thoroughly revised the revision and have addressed every suggestion and comment in detail. We believe that the quality of the manuscript has been greatly improved by these revisions.
Reviewer 1
The study is well structured and understandable and very practical. It is important to investigate the fabrication of aligners using new methods to generate the best results for patients.
In the introduction it is mentioned that the models are manufactured by lithography based AM technologies (DLP, SLA, jetting) nowadays. (page 2 76-77) and that there is a great uncertainty between published data. It seems that there is no clear statement regarding the accuracies of dental models in the literature or very controversial data has been published.
In the literature in the introduction, there are no references to the stability of models fabricated using lithography-based AM technologies after thermoforming as in the submitted study. To ensure comparability with the most commonly used technologies for the fabrication of models for aligner fabrication, at least one Vat lithography-based AM technology and polymer jetting based AM technology should be included in the recent study design. Additive manufacturing of models for aligner fabrication is already standard today and must be considered when new materials and technologies are investigated as alternatives.
Why were only filaments from Renfert used? There are other filaments on the market that are designed for this indication (Filadental; Weithas; Dental 3D Agency).
I recommend making up the required groups and then resubmitting to Materials.
Authors comments: Thank you very much for your reasonable recommendation. We agree with you that it would be appropriate to include further filaments of other manufacturers in the study – among others to avoid bias of the study outcomes. However, we are worried that the study would then become too extensive and the presentation of the results too confusing. For this reason, we would like to avoid expanding the study even further at this point in order to maintain the focus, but at the same time we have included this aspect in the discussion as a limitation of the study.
Revised text (p. 14, ll. 433-441): Overall, it would be reasonable to verify the present observations via polymer analytics using FTIR or GPC analysis as well as by examining further filaments from other manufacturers intended for the fabrication of models for aligners. Both, the limited information available from the manufacturer for the investigated polymers and the fact that the present investigation examined only two filaments from one and the same manufacturer must be considered a limitation of the study. Both limitations should be considered in future studies. Furthermore, future study designs could benefit from considering alternative, already established additive manufacturing methods of models such Vat lithography-based AM technology and polymer jetting based AM technology.
While reading, I noticed terms that should be checked again, such as:
Fused Filament Fabrication (page 2 80); when checking, the "ISO/ASTM 52900 - Terminology for AM - General Principles - Terminology" should be taken into account and correct terms should be used.
The term "Fused Deposition Modeling" is not spelled correctly (page 2 80) and is a trademark term of Stratasys and should therefore be avoided in scientific publications. Please change throughout the manuscript.
Authors comments: Thank you very much for your thoughtful comment. The manuscript was revised accordingly; the term “Fused Deposition Modeling” has been replaced by the term “Fused Filament Fabrication” or by the corresponding abbreviation “FFF” throughout the manuscript.
Revised text: Please refer to the current manuscript for all revisions to the term “FDM”.
Furthermore, could the descriptive statistic in Table 3 be formatted as a graph? This would make it easier for the reader to visually examine the comparisons. Examples can be found in Son, K.; Lee, W.-S.; Lee, K.-B. Effect of Different Software Programs on the Accuracy of Dental Scanner Using Three-Dimensional Analysis. Int. J. Environ. Res. Public Health 2021, 18, 8449. https:// doi.org/10.3390/ijerph18168449.
Authors comments: Thank you very much for this important und very good suggestion. We have tried to implement it. However, since we tested 42 groups, we find it very confusing to present the descriptive statistics within a diagram and have stayed with the table.
Revised text: -
Reviewer 2 Report
The paper proposes an interesting application of FDM process with the intent of replacing costly and weak plaster models of maxilla and mandible. The authors compared different filament materials (PTA and PETG), different thermoforming foils and cycles, different models of jaw, and different printing parameters (number of loops); revealing interesting effects and conclusions on FDM models' stability. The paper is well written and there are no big modification to perform. Anyway, before to be accepted for publication, in the opinion of the reviewer, a minor revision is necessary.
In the following, the authors can find the suggestions of the reviewer:
- the caption of Table 1 cited that also the expiration is presented, but it is not. Please remove "and expiration" from the caption;
- the page numbers are wrong from the horizontally oriented Table 3. In particular the landscape pages numbering restarted from page 1, and the following portrait pages continued with the new numbering. This can be a bit confusing, please fix the page numbering.
- the caption of figure 8 indicates mandibulae as A and maxillae as B, but no letters in the picture are visible. Please fix it by removing the A and B letters in the caption, or by adding A and B letters in the picture.
- the second item of the conclusion reports that Renfert PLA HT is less stable than Renfert PLA HT (itself). The reviewer thinks that this is a refuse since from the analysis a better stability of SIMPLEX is understandable. Please fix this refuse.
Author Response
Revision of manuscript: materials-2471594
Title: Dimensional stability and reproducibility of varying FDM models for aligners in comparison to plaster models
Dear Editor,
We thank you and the reviewer for your suggestions, insightful comments and corrections. We appreciate your efforts to evaluate our manuscript and providing us with the opportunity to make the following revisions. We have thoroughly revised the revision and have addressed every suggestion and comment in detail. We believe that the quality of the manuscript has been greatly improved by these revisions.
Reviewer 2
Comments and Suggestions for Authors
The paper proposes an interesting application of FDM process with the intent of replacing costly and weak plaster models of maxilla and mandible. The authors compared different filament materials (PTA and PETG), different thermoforming foils and cycles, different models of jaw, and different printing parameters (number of loops); revealing interesting effects and conclusions on FDM models' stability. The paper is well written and there are no big modification to perform. Anyway, before to be accepted for publication, in the opinion of the reviewer, a minor revision is necessary.
In the following, the authors can find the suggestions of the reviewer:
- The caption of Table 1 cited that also the expiration is presented, but it is not. Please remove "and expiration" from the caption;
Authors comments: Thank you very much for the advice. The caption of Table 1 was revised accordingly.
Revised text (p. 4, ll. 155-156): “Table 1: List of used materials sort by type of material, product name, manufacturer and LOT number”
- The page numbers are wrong from the horizontally oriented Table 3. In particular the landscape pages numbering restarted from page 1, and the following portrait pages continued with the new numbering. This can be a bit confusing, please fix the page numbering.
Authors comments: Thank you very much for pointing this out. Numbering of pages has been revised and is henceforth continuous throughout the manuscript.
Revised text: Please find the changes in the revised manuscript.
- The caption of figure 8 indicates mandibulae as A and maxillae as B, but no letters in the picture are visible. Please fix it by removing the A and B letters in the caption, or by adding A and B letters in the picture.
Authors comments: Thank you very much for this advice. It was noticed that the figures in the manuscript were mixed up. The allocation of the figures has now been corrected in the manuscript as part of the revision.
Revised text: Please find the changes in the revised manuscript.
- The second item of the conclusion reports that Renfert PLA HT is less stable than Renfert PLA HT (itself). The reviewer thinks that this is a refuse since from the analysis a better stability of SIMPLEX is understandable. Please fix this refuse.
Authors comments: Thank you very much for pointing out the mistake. The conclusion has been revised accordingly.
Revised text (p. 16, ll. 541-544): “Both, the filament used, and the loop counts have a significant impact on the dimensional stability with Renfert PLA HT being less stable than SIMPLEX aligner model filament and with a loop count of 3 and 4 showing comparable dimensional stability for Simplex Aligner.”
Reviewer 3 Report
(1)The difference of the five polymers may be mentioned.
The FTIR characterizations are suggested to be taken for the five polymers used, which can directly show the difference of these five polymers.
(2) These polymers used are very popular. The authors may have to provide some discussion about previous study.
(3) The samples surface is suggested to be shown, which can be directly measured via an optical microscope or a fluorescence microscope (The samples can be coated with dye). SEM may be not required.
Minor editing of English language required.
Author Response
Revision of manuscript: materials-2471594
Title: Dimensional stability and reproducibility of varying FDM models for aligners in comparison to plaster models
Dear Editor,
We thank you and the reviewer for your suggestions, insightful comments and corrections. We appreciate your efforts to evaluate our manuscript and providing us with the opportunity to make the following revisions. We have thoroughly revised the revision and have addressed every suggestion and comment in detail. We believe that the quality of the manuscript has been greatly improved by these revisions.
Reviewer 3
Comments and Suggestions for Authors
- The difference of the five polymers may be mentioned.
The FTIR characterizations are suggested to be taken for the five polymers used, which can directly show the difference of these five polymers.
Authors comments: Thank you very much for the valuable suggestion to identify the differences between the polymers via FTIR spectroscopy. We appreciate the proposal and think this is a useful approach, however, our current infrastructure does not allow access to an FTIR spectrometer. For this reason, we currently have to rely on the information that is available to us from the manufacturers for the polymers under investigation. Nevertheless, the objection to characterize differences between the polymers via FTIR analysis in order to substantiate the observations of the investigation more soundly was included in the discussion of the present investigation.
Revised text (p. 16, ll. 474-480): Overall, it would be reasonable to verify these observations via polymer analytics using FTIR or GPC analysis as well as by examining further filaments from other manufacturers intended for the fabrication of models for aligners. Both, the limited information available from the manufacturer for the investigated polymers and the fact that the present investigation examined only two filaments from one and the same manufacturer must be considered a limitation of the study. Both limitations should be considered in future studies.
- These polymers used are very popular. The authors may have to provide some discussion about previous study.
Authors comments: Thank you for this comment. We agree that it is useful to discuss the latest findings on the investigated filaments from the scientific literature. The discussion of the manuscript has been revised accordingly.
Revised text (p. 15, ll. 431-433; p. 15-16, ll. 438-471): In contrast, the SIMPLEX aligner model filament consists of glycol-modified polyethylene terephthalate PETG, which is one of the most used thermoplastics in 3D printing due to its high impact resistance and ductility.
[…]
A recent investigation analyzed the dynamic mechanical and thermal properties of PETG aligners after thermoforming and artificial aging and demonstrated that thermoforming had a more prominent role than aging in diminishing of thermomechanical properties of the thermoplastic polymer [17]. Both, flexural modulus, and hardness revealed a significant decrease by thermoforming just as the dynamic glass transition temperature. Latter observation was said to indicate the attenuation of thermomechanical properties after this process. In general, it was noted that some recent scientific investigations focused on the impact of temperature on the printability and the related material properties of PETG [18-20]. Guessasma et al. determined a glass transition temperature of 77°C for PETG, which is consistent with the information on the material in the present study and indicated that it can only be printed within a limited range of printing temperatures requiring temperature above 230°C [18]. Furthermore, they found that FFF processing of PETG resulted in a significant reduction in elongation at break and a significant loss of both stiffness (up to 40%) and tensile strength (up to 40%) of PETG. At a printing temperature of 250°C, a slight improvement in the tensile strength of PETG was observed, but the loss in both toughness and tensile strength still accounts for one-third of the raw material [18]. These findings are in general agreement with the research results of Valvez et al. [20]. Here, the optimum nozzle temperature was identified slightly higher, at 265°C in further dependence on the printing speed and the layer thickness, to achieve the highest possible bending properties for neat PETG. However, again the study outcomes indicated the temperature being one of the most relevant parameters determining the mechanical properties of PETG when processed by FFF technology. Furthermore, the research group of Hsueh et al. also investigated the influence of printing parameters (temperature and speed) on the thermal and mechanical properties of PETG and PLA using FFF technology [19]. Both polymers are also subject of the present investigation. Hsueh et al. showed that the mechanical properties of PLA and PETG increased with increasing printing temperatures. The same observation was found for PLA with increasing printing speed, while for PETG, a decrease in mechanical properties was identified a printing speed increased. However, in relation to the present study, the most interesting finding showed that PLA has the higher mechanical properties compared to PETG but is less resistant to thermal deformation [19], which is in perfect agreement with the observations of the present investigation. In general, the finding from the current literature that the pressure parameters have a significant impact on the mechanical and thermal properties of FFF filaments is consistent with the observations of the present investigation. Here, too, it has been shown that it is not the FFF filament itself, but the related printing parameters (loop counts) that have the greatest influence on dimensional stability.
- The samples surface is suggested to be shown, which can be directly measured via an optical microscope or a fluorescence microscope (The samples can be coated with dye). SEM may be not required.
Authors comments: Thank you for the suggestion to include pictures of the specimen surfaces in the presentation of the results. The authors decided to omit the presentation of the specimen surfaces as they do not add any value to the results already presented. From the authors point of view, no relevant differences in the specimen surfaces were observed that need to be presented or discussed.
Revised text: n/a
Comments on the Quality of English Language
Minor editing of English language required.
Round 2
Reviewer 1 Report
Thanks for the corrections and further discussion!